# An Updated Review on Monkeypox Viral Disease: Emphasis on Genomic Diversity

**DOI:** 10.3390/biomedicines11071832

**Published:** 2023-06-26

**Authors:** Ali A. Rabaan, Nada A. Alasiri, Mohammed Aljeldah, Abeer N. Alshukairiis, Zainab AlMusa, Wadha A. Alfouzan, Abdulmonem A. Abuzaid, Aref A. Alamri, Hani M. Al-Afghani, Nadira Al-baghli, Nawal Alqahtani, Nadia Al-baghli, Mashahed Y. Almoutawa, Maha Mahmoud Alawi, Mohammed Alabdullah, Neda A. Al Bati, Abdulmonem A. Alsaleh, Huseyin Tombuloglu, Kovy Arteaga-Livias, Tareq Al-Ahdal, Mohammed Garout, Mohd Imran

**Affiliations:** 1Molecular Diagnostic Laboratory, Johns Hopkins Aramco Healthcare, Dhahran 31311, Saudi Arabia; 2College of Medicine, Alfaisal University, Riyadh 11533, Saudi Arabia; 3Department of Public Health and Nutrition, The University of Haripur, Haripur 22610, Pakistan; 4Monitoring and Risk Assessment Department, Saudi Food and Drug Authority, Riyadh 13513, Saudi Arabia; 5Department of Clinical Laboratory Sciences, College of Applied Medical Sciences, University of Hafr Al Batin, Hafr Al Batin 39831, Saudi Arabia; 6Department of Medicine, King Faisal Specialist Hospital and Research Center, Jeddah 21499, Saudi Arabia; 7Infectious Disease Section, Internal Medicine Department, King Fahad Specialist Hospital, Dammam 32253, Saudi Arabia; 8Department of Microbiology, Faculty of Medicine, Kuwait University, Safat 13110, Kuwait; 9Microbiology Unit, Department of Laboratories, Farwania Hospital, Farwania 85000, Kuwait; 10Medical Microbiology Department, Security Forces Hospital Programme, Dammam 32314, Saudi Arabia; 11Molecular Microbiology and Cytogenetics Department, Riyadh Regional Laboratory, Riyadh 11425, Saudi Arabia; 12Laboratory Department, Security Forces Hospital, Makkah 24269, Saudi Arabia; 13iGene Center for Research and Training, Jeddah 2022, Saudi Arabia; 14Directorate of Public Health, Dammam Network, Eastern Health Cluster, Dammam 31444, Saudi Arabia; 15Directorate of Health Affairs, Al-Ahsa Health Cluster, Ministry of Health, Al-Ahsa 31982, Saudi Arabia; 16Primary Healthcare, Qatif Health Network, Eastern Health Cluster, Safwa 32833, Saudi Arabia; 17Department of Medical Microbiology and Parasitology, Faculty of Medicine, King Abdulaziz University Hospital, Jeddah 22254, Saudi Arabia; 18Infection Control and Environmental Health Unit, King Abdulaziz University Hospital, Jeddah 22254, Saudi Arabia; 19Department of Infectious Diseases, Almoosa Specialist Hospital, Al Mubarraz 36342, Saudi Arabia; 20Medical and Clinical Affairs, Rural Health Network, Eastern Health Cluster, Dammam 31444, Saudi Arabia; 21Clinical Laboratory Science Department, Mohammed Al-Mana College for Medical Sciences, Dammam 34222, Saudi Arabia; 22Department of Genetics Research, Institute for Research and Medical Consultations (IRMC), Imam Abdulrahman Bin Faisal University, Dammam 34221, Saudi Arabia; 23Escuela de Medicina-Filial Ica, Universidad Privada San Juan Bautista, Ica 11000, Peru; 24Escuela de Medicina, Universidad Nacional Hermilio Valdizán, Huanuco 10000, Peru; 25Research Associate, Institute of Global Health, Heidelberg University, Neuenheimerfeld130/3, 69120 Heidelberg, Germany; 26Department of Community Medicine and Health Care for Pilgrims, Faculty of Medicine, Umm Al-Qura University, Makkah 21955, Saudi Arabia; 27Department of Pharmaceutical Chemistry, Faculty of Pharmacy, Northern Border University, Rafha 91911, Saudi Arabia

**Keywords:** monkeypox, genome, mutations, molecular, treatment

## Abstract

Monkeypox virus has remained the most virulent poxvirus since the elimination of smallpox approximately 41 years ago, with distribution mostly in Central and West Africa. Monkeypox (Mpox) in humans is a zoonotically transferred disease that results in a smallpox-like disease. It was first diagnosed in 1970 in the Democratic Republic of the Congo (DRC), and the disease has spread over West and Central Africa. The purpose of this review was to give an up-to-date, thorough, and timely overview on the genomic diversity and evolution of a re-emerging infectious disease. The genetic profile of Mpox may also be helpful in targeting new therapeutic options based on genes, mutations, and phylogeny. Mpox has become a major threat to global health security, necessitating a quick response by virologists, veterinarians, public health professionals, doctors, and researchers to create high-efficiency diagnostic tests, vaccinations, antivirals, and other infection control techniques. The emergence of epidemics outside of Africa emphasizes the disease’s global significance. Increased monitoring and identification of Mpox cases are critical tools for obtaining a better knowledge of the ever-changing epidemiology of this disease.

## 1. Introduction

Monkeypox (Mpox) is a viral zoonotic disease, transferred to humans from animals, but once it is transmitted to humans, human-to-human transfer is also possible [1,2]. The symptoms of Mpox are quite similar to those that were observed in the past in individuals who were suffering from smallpox, despite the fact that Mpox is clinically less severe. It is brought about by the monkeypox virus (MPXV), which is classified as an orthopoxvirus (OPVX) and is a member of the Poxviridae family of viruses [3]. Poxviridae family and OPVX genus viruses consist of various viruses known to infect humans, including monkeypox virus (MPXV), vaccinia virus (VACV), cowpox virus (CPXV), and variola virus (VARV) [4,5]. Genes involved in the determination of host range and pathogenicity can be found at the virus’s changeable terminal ends, but the genomes of these viruses have highly conserved middle sections that code for replication and assembly machinery [6,7].

The Congo Basin clade (Central African) (clade I) and West African clade (clade IIa and IIb) are the subtypes of the MPXV [8]. These subtypes were found in Africa. The virus that causes Mpox was first identified during 1958 in the animals (monkeys) in a research laboratory in Denmark [9]. In 1970, the first human case was found in the Democratic Republic of the Congo (DRC), as described by Ladnyj et al. [10], which was discovered in a child. This virus (MPXV) can be transferred from one person to another by lesions, respiratory droplets, body fluids, and contaminated things such as bedding that come into intimate contact with the infected person [11]. In most cases, the incubation period for Mpox lasts 6–13 days, but this time frame can last anywhere in the range of 5–21 days [12,13,14].

Since the reporting of the first case, this disease has grown endemic in the DRC, and it is now prevalent in other African countries, mostly those located in Central and West Africa [15,16]. The first Mpox case outside African territory was reported in 2003, and 2019 was the year that had the most simultaneous instances [8,17].

Mpox developed capabilities with catastrophic implications due to its high adaptability to humans [18]. The possibility of new reservoirs of Mpox being established in regions other than Africa is a cause for concern regarding the importation of MPXV by infected vertebrates. In point of fact, it has been discovered that ground squirrels in the United States are vulnerable to infection, which suggests that other rodent species all over the world might also be vulnerable [19,20]. Even minute genetic shifts could make it easier for a disease to adapt to humans as hosts. The likelihood of this happening is always good for the kind of pathogens that have an average rate of transmission [21,22,23]. The capacity to spread quickly and effectively from human to human could facilitate the expansion of the disease’s presence in human populations into previously unexplored areas. As a result, active disease surveillance needs to be maintained so that MPXV may be monitored for changes that are compatible with its increased adaption to humans [24,25]. Discovering the true geographic spread of this virus requires continued intensive surveillance in the Sankuru District, as well as expansion of that surveillance to all other places where the virus is known to circulate or where it is anticipated to circulate [26]. In light of the apparent rapid evolution of this virus, health authorities in areas that are not yet afflicted by it must be on red alert and actively ready to take immediate action in the event that suspected or confirmed instances of the disease are found in humans [27,28] (Figure 1).

The definition of cases is not per any specific standard in all given sources. The suspected cases are those cases in which a sudden high-grade fever leads to vesicular pustule eruption abundantly present on the face, hands, palms, feet, and soles or minimum five smallpox scabs [29,30]. A clinical sign differentiating Mpox from smallpox is lymphadenopathy. On the other hand, the new Mpox outbreak clinical presentation has been atypical as compared to previously documented reports in Mpox endemic areas of Africa. Anal pain and bleeding, genital or only perineal/perianal lesion, and absence of prodromal period or constitutional symptoms appearing after the lesion are features of the new Mpox outbreak. Probable cases are those cases that are identified based on epidemiological character and are usually without laboratory confirmation [31]. The possible cases are noted usually with vascular rash and fever history [32]. Finally, the confirmed cases are any cases with laboratory results confirmation, most likely by PCR or antibody testing [33,34] (Figure 2).

### Genetic Variability

While OPVXs are genetically and antigenically identical, they have varied host range and pathogenicity features. An OPVX’s evolutionary trajectory can be influenced by a host species’ selective pressure [35,36]. Virus evolution may have been sped up by the loss of genes, particularly near the ends of genomes, as has been hypothesized. The largest sequenced OPVX genome (220 kb) has 223 open reading frames (ORFs) and prevalent hosts, including rodents, people, cats, dogs, and voles [16]. Cowpox virus only causes mild infection in humans. Conversely, VARV, the smallpox causative agent, is extremely deadly, with a more than 30% fatality rate. It has the smallest genome of any naturally existing OPVX [37].

The DNA genome of MPXV is approximately 197 kilobases long and contains around 190 ORFs that are longer than 180 nucleotides. The coding region sequence at MPXV nucleotide locations 56,000–120,000 is extremely conserved, just like it is in all OPVXs [20]. The majority of VACV gene homologs discovered in the distal end of the MPXV genomes are involved in immune modulation, and the majority of these homologs are either anticipated to influence host range determination and pathogenicity or confirmed to have such an influence. In contrast to VARV, which does not possess any ORF in the inverted terminal repeats (ITRs) area, MPXV possesses at least four ORF in the ITRs region [38].

Overall, a polymorphism was found in the ITR region across the MPXV alignments. This polymorphism contained 12 different variants [9]. In total, 4 of the full genome sequences, which accounted for 17.4% of the total, displayed significant instability of the genome just before ITR upstream. Positions 189,820 and 190,444 in this specific batch of data experienced a loss of 625 base pairs. Both MPV-Z-N2R and the first 103 base pairs of the MPV-Z-N3R OPVX class I major histocompatibility complex-like protein have been completely deleted as a result of this deletion. The function of the MPV-Z-N2R protein is unknown, and neither the VARV nor the West African MPXV genomes include a gene that is related to it. An example of a secreted protein is the OXPV MHC Class I-like Protein (OMCP), which binds to natural killer group 2, member D (NKG2D), and stops natural killer cells from killing cells via NKG2D’s mediating action. We were able to discover this loss in six more genomes by using a standard PCR that we developed to analyze the deletion. There was no indication of a homogenous population in either the data of sequences for the isolates that may have had a deletion or in the electrophoresis of the Amplicon Sanger sequencing product for the sample that was put through its paces [39].

Although models of host transition might anticipate genomic changes, the association between secondary transmission and gene loss pattern might imply that MPXV is adjusting for effective replication in an unique ecological niche represented by humans [14,40]. It is also possible that the association between OMCP transmissibility and gene loss is co-incidental. This is due to the fact that numerous other factors, such as vaccination status and human incursion into reservoir habitat types, could also clarify growing human spread and variant introduction frequency. Vaccination status and land encroachment of humans on reservoir habitats are examples of such factors [41]. We are unable to pin-point the exact source of the assessed variability due to the lack of information regarding the historical emergence of OPVXs and the absence of sequencing data for MPXV reservoir isolates. In any case, it was predicted that the four lineages are active in the population of the reservoir and later on intrude into the human population after direct contact with those host reservoirs [37] (Figure 3).

## 2. Geographical Range and Progressive Epidemiology of MPXV

The MPXV has traditionally been found in the tropical central and Western African rainforests of countries, most notably the DRC; however, there is a possibility that the virus’ distribution is spreading [42,43]. Since 2017, there has been a large outbreak of Mpox in Nigeria, linked with cases in 2018–2019 outside endemic areas of Africa and probably with the new outbreak starting in May 2022 [44]. The MPXV entered the United States in 2003, and it was associated with a shipment of approximately 800 small mammals from Ghana that contained 762 African rodents, including squirrels, rats, dormice, and porcupines [45]. In 2005, an epidemic of the disease was detected in Sudan [46]. The reservoirs of the MPXV in the wild are most likely African squirrel types (*Funisciurus* and *Heliosciurus* sp.), other rodents, and possibly monkeys. The CPVX is endemic throughout Europe and a few western states in the countries that were a part of the former Soviet Union. The virus has a very diversified genetic makeup [44,47]. It has been suggested that rodents, including voles, wood mice, and rats, serve as reservoirs for the cowpox virus, while cattle, zoo animals, and domestic cats serve as incidental hosts [48,49] (Table 1).

Since the 1970s, the number of reported cases of human Mpox has been steadily climbing, with the DRC showing the most significant increase [58,59]. The median age of presentation is currently 21 years, which is a significant increase from the 1970s, when it was just 4 [60]. In a previous study, there was a significant difference in the case fatality rate between clades, with Central African cases having a rate of 10.6% (95%, CI: 8.4–13.3%) and West African cases having a rate of 3.6% [61]. The overall case fatality rate was 8.7%. Since 2003, sporadic outbreaks have been caused by the spread of the disease outside of Africa as a result of imports and travel [62,63]. Behaviors that put a person at risk for contracting Mpox include having interactions or engaging in activities with infected animals or people. According to the findings of our analysis, the number of cases of CPVX has been rising, particularly in the DRC, where this disease was endemic, the disease has spread to neighboring countries, and the median age of patients has decreased from young adults to young children [64,65,66]. Such findings may have some bearing on the decision to stop vaccinating against smallpox, which offered a kind of cross-protection against MPXV but ultimately led to an increase in the spread of the disease from human to human. The worldwide significance of the disease has been brought into focus by the advent of outbreaks in regions other than Africa [67,68,69].

From 1970 to 2019, a total of 1347 confirmed and over 28000 suspected cases were registered in the DRC [70,71]. The DRC is the country with the highest number of incidences of Mpox, and no other country has consistently reported instances of Mpox during the past half-century [72]. The primary type of case that was observed after the year 2000 was one in which the patient’s illness was only suspected rather than one in which the patient’s illness was confirmed, probable, or possible [73,74]. More recently, between the months of January and September of the year 2020, 4594 additional suspicious patients were observed and noticed in the DRC. Nigeria is the 2nd most highly impacted area, with 181 probable and confirmed cases associated with the outbreaks that began in the 9th month of 2017 [75]. It should be noted that the Center for Disease Control (CDC) report for Nigeria lists 183 instances; however, two of those cases originated in Nigeria and were diagnosed in Singapore and Israel [76]. These two cases were designated travel-related occurrences for their respective countries. The 183 cases that were reported by the Nigerian CDC do not include the cases that occurred in the United Kingdom (UK) and originated in Nigeria [77]. In 2008, two separately imported cases of human Mpox in the UK were reported. Both had travelled in southern Nigeria before coming to the UK [77]. The DRC (*n* = 97) and the Central African Republic (*n* = 69) are the third and fourth most afflicted nations regarding potential, confirmed, and probable cases of MPXV, respectively [78]. Over the course of the preceding half-century, the total number of suspected and confirmed cases of Mpox in each of the remaining African nations was less than 20 [79,80,81,82] (Table 2).

Smallpox epidemiology, with causative agent, the variola OPVX, has been understood by in-depth investigations that were carried out after the eradication campaign was successfully completed [84,85]. Inhaling a huge airborne respiratory droplet of a virus that can cause infection was the primary mode of human-to-human transmission of the variola virus. Transmission often requires extended face-to-face or other close contact; nevertheless, there have been reports of aerial transmission over larger distances [86,87]. Transmission could potentially have occurred via fomite or through contact with infected things originating from the rashes. The smallpox eradication campaign resulted in the collection of aggregate data that suggest the rate of secondary attack was 58.4% in unvaccinated households. These figures were derived from a comparison of both groups [88,89]. Under the conditions of current experimental and analytical findings, the airborne transmission route must be considered as a possible transmission mode. The findings and analysis with aerosol dynamics showed that aerosols carrying MPXV could be present in environments where patients have resided and that airborne transmission of MPXV can occur [77,90]. For variola, the case fatality rate majorly varied depending on the manifestation of the disease; nonetheless, aggregate case fatality rates ranging from 10% to 30% have been observed over a number of outbreaks [91,92]. The severity of the condition was found to have a correlation with the amount of rash that was present, and the condition was also found to be adverse in children and in women during pregnancy [93,94] (Table 3).

The epidemiology of Mpox is significantly more complicated [96]. The virus is zoonotic, and it has been found that there are three genetically separate virus clades, each of which appears to have distinct clinical and epidemiologic features [97,98,99]. There were 122 cases in total in seroprevalence surveys, suggesting that subclinical infections may have occurred in up to 28% of the patients who had close proximity with animals in some communities. This could be a contributing factor to the sustained rarity of transmission between humans in households and other close-contact situations [86,100,101]. It wasn’t until 1970 that researchers found evidence of infections in humans of Central and Western Africa. In the country that is now known as the DRC, which was formerly known as Zaire, researchers found that transmission between humans of MPXV was significantly less common than the transmission of smallpox [40,102]. The computed secondary attack rate in unvaccinated contacts of Mpox cases was 9.3%, whereas the secondary attack rate in unvaccinated contacts of smallpox patients ranged from 37% to 88% [103]. Previous vaccination against smallpox, which could have been given anywhere from 3 to 19 years in the past, was found to be 85% protective [104,105]. Only 28% of cases were attributed to the spread of the disease from one person to another, while the vast majority of documented individuals contracted the disease from assumed animal exposure [57]. A case fatality rate of roughly 10% was seen in unvaccinated individuals, and children younger than 5 years old accounted for the bulk of fatalities and the most severe illness symptoms that were recorded [106,107]. Cases of Mpox in Africa were comparatively few before the 1980s, following smallpox vaccination discontinuation worldwide, and in West and Central Africa, the risk of human Mpox outbreaks has been growing every year since that date. The smallpox vaccination provides cross-immunity against Mpox. Since 1977, the end of smallpox vaccination has resulted in a decreased immunity and in an increased population susceptible to Mpox. 

## 3. Genome Organization, Replication Cycle, and Morphology

The genome of MPXV is a linear double-stranded DNA genome that is approximately 197 kilobases in size [41,108]. In spite of the fact that the MPXV genome consists of DNA, its whole life cycle takes place within the cytoplasm of cells that have been infected [109]. The MPXV genome encodes all of the proteins that are necessary for replication of DNA, virion assembly, transcription, and egress. The hairpin loops with tandem repeats in ORFs combined to form ITRs [110,111]. This includes all of the proteins [112] (Figure 4).

It is thought that IMV is released during cellular breakdown and EEV is released from the cells through interaction with actin tails, which contributes to the particular virus’s fast and extensive area diffusion inside an infected host [113,114]. These two things happen at the same time. Despite the fact that the afore-mentioned characteristic is specific to VACVs, it is highly likely that these properties are shared by all OPXV. Cell-associated virions (CEVs), on the other hand, are produced after the microtubule-mediated transportation of an enveloped virus inside cellular peripheries [115,116]. During this process, the outmost plasma membrane of the intracellular enveloped virus (IEV) binds with the cellular plasma membrane, which remains bonded to the surfaces of cells [117]. This ends in the formation of cell-associated virions (CEVs). Cell-associated virions are the primary agents of cell-to-cell transmission, as described in [118]. When IMV is encased in a double membrane that originates from either the trans golgi network or early endosomal component, IEV is produced [119,120]. However, separate from the process of IEV exocytosis, another way that EEV can be formed is through the IMV budding through the plasma cellular membrane. This is an alternative pathway for the production of EEV. It has been reported that the virion morphogenesis process in the prototype VACV can go awry, leading to the formation of dense particles that are not infectious [121]. However, this behavior has not been observed in MPXV [122]. Due to a truncation in the ATIP gene, MPXV sequesters IMVs into A-type inclusion (ATI) or does not create ATIs. This is in contrast to certain strains of CPXV, in which IMV is occluded within ATIs. In addition, MPXV does not produce ATI or sequester IMV into ATIs [123,124] (Figure 5).

The MPXV morphology reveals that the virions are brick-shaped or ovoid particles that are enveloped by a lipoprotein, which is geometrically corrugated into the outer membrane [125,126]. These virions share the same physical properties as other OPVXs. It is known that the size range of MPXV is 200–250 nm. The plasma membrane plays a role of protection of the membrane bond as well as a densely packed core [127]. The core is described as biconcave, and it has lateral bodies on both sides because of an anomaly caused by the fixation technique used in electron microscopy, [54,128].

### Clades of Monkeypox

There are substantial ramifications associated with the methodical gathering of surveillance data for efficient disease control and prevention [129]. Estimating the burden of disease, monitoring changes in disease occurrence, determining geographic spread, identifying high-risk populations and other health concerns, and informing resource allocation are all things that can be accomplished with the help of surveillance systems that are in good working order. As a result, one of the most important things that needs to be carried out for public health is an analysis of the efficiency of monitoring systems and how they develop over time [130].

The existence of three distinct MPXV clades was established using genomic sequencing of isolates originating from the United States of America, Western Africa, and Central Africa [103]. The ability to anticipate proteins in the virus that could be responsible for the different pathogenicity between MPXV clades was made possible by conducting comparative analysis between open reading frames of MPXV. The prevention and control of Mpox can be improved with a better understanding of the molecular etiology and clinical and epidemiological characteristics of MPXV [19,109].

Comparisons between the genomes of these clades of MPXV and related OPXV have shown that despite the fact that these viruses are closely related to one another, they contain a number of genome region variations that are distinguished by insertions, high mutation rates, gene truncations and deletions. These regions have been designated [9,131].

## 4. Phylogenetic Analysis of MPXV

The genomic sequences of Congo Basin and West African derived strains of MPXV found an overall 99% identity of nucleotides within geographical areas and only 95% nucleic acid identity over all geographical clusters. It is important to note that there was just a single nucleotide variation between MPXV-USA-2003-039 (human) and MPXV-USA-2003-044 (prairie dog) throughout the first two cycles of viral transmissions, possibly originating from the same source [132].

The majority of ORFs that had pi values of greater than six were segments of OPXV isoforms. Analyses of phylogenetics, using highest compatibility and similarity in methodology, were performed using the four different geographical MPXV genome sequences, including the earlier described MPXV-ZAI-1996-016, and were deeply embedded with VACV strain Copenhagen and Grishak (CPXV-GRI) a strain of cowpox virus. These analyses were carried out using MPXV-ZAI-1996-016 as the outgroup (VACV-COP). 

Comparison of genome sequence and alignments between MPXV clades and some other similar OPXVs allowed identifying two target genomic regions (R) positioned at the 5’ end and 3’ end of the genome. These R areas are located on opposite ends of the genome. The DNA sequences of MPXV and related OPXVs were retrieved from Genbank databank and examined with the help of several bioinformatics programs. The mutation rates and the occurrence of large-scale evolutionary processes, such as genomic re-arrangement, truncation, inversion, deletion, and insertion, were taken into consideration while selecting genomic regions for study. In addition, each genomic area was chosen based on the degree of divergence it shared with other viruses in the same family, as well as the number of known virulence genes it contained. As a result of this study, two genetic areas have been chosen for further thorough investigation. The MPXV-R1, MPXV-R1/R2, and MPXV-R2 recombinant virus species all have combined genomic areas or individual deletions and were created by a recombination process. Plaque purification, titration, and testing of recombinant viruses using cell culture technique and in vivo imaging were all performed on the viruses [133,134]. (Figure 6).

It is feasible to further boost the resolving power of these phylogenetic assessments by using datasets that contain greater numbers of shared orthologous genes. This would permit a more extensive investigation of these viruses’ evolutionary history. Although only 20 gene families were able to have their sequences matched in a way that was completely unambiguous for analysis support that was given previously, a minimum of 1 fewer than 50 genes significantly exhibit some level of sequence similarity across the complete family. A core set of 174 genes is almost identical in all OPXVs, and all strains of every species of virus related to the OPXV genus share some portion of the 214 genes that are found in CPXV. An estimated ninety gene families within the OPXV genus exhibit significant homologous sequences across the subspecies [131].

### Nigerian Phylogeny

Five Nigerian isolates were from the 2017 Rivers State outbreak [135], while the genomes of two other Nigerian isolates (Monkeypox-W-Nigeria and MPXV-SE-Nigerian) prevailed in 1978 during early outbreaks (Oyo State) and 1971 (Abia State) [136]. During September 2018, the genome of the MPXV isolate was transported to Israel (MPXV Israel) from a Port Harcourt, Rivers State, resident who returned to Israel seven days after disposing of two rodent carcasses in his Port Harcourt apartment. MAFFT version 7 was used to align all of the sequences. A phylogenetic tree was built from the aligned sequences using the MEGA X software’s Maximum Likelihood (ML) technique [137]. The WA clade comprised eight Nigerian isolated strains and also some from Liberia and the United States delivered from Ghana. Furthermore, the West African clade was further divided into two groups, with Nigerian isolates forming a subgroup different from a second subcluster made up of Liberian and American isolates. MPXV-Nigeria W and the Nigerian SE-1971 are genetically distinct from other previous isolates. The latest Nigerian isolates’ monophyly with the Israel exported isolate suggests that the virus may have arisen from the same pool of infection [63]. This matches a previous phylogenetic analysis of Bayesian MPXV transferred from Nigeria to other regions of the world. Isolates from Sudan, the Congo, Cameroon, and Gabon make up the Congo Basin clade. Despite the fact that Gabon and Cameroon are West African countries, MPXV resolved into the Congo Basin clade, which was isolated from these two countries. A previously published paper’s MPXV tree topology agrees with many other published results [63].

## 5. Conclusions

The declining population immunity caused by the discontinuance of smallpox vaccination has created a favorable environment for the comeback of MPXV. This is evidenced by a rise in the confirmed cases and specific age of those with Mpox, as well as the reemergence of outbreaks in some regions after a 30–40-year gap. The expansion of instances outside of Africa also highlights the disease’s potential for global importance and geographic diffusion. Concern about human-to-human transmission affects not just family members but also caretakers for the patients. The public health significance of the Mpox epidemic should not be understated given the contemporary context of pandemic risks. Understanding the dynamic epidemiology of this emerging disease requires increased global monitoring and the early identification of Mpox cases. To maintain influence and support informed decision making regarding the disruptions of MPXV infections in humans, potential future outbreak responses, diagnostic test deployment, and even prospective outbreak-related decisions regarding vaccination and medications, more research into the involvement of these different MPXV clades in human disease is required.

## Figures and Tables

**Figure 1 biomedicines-11-01832-f001:**
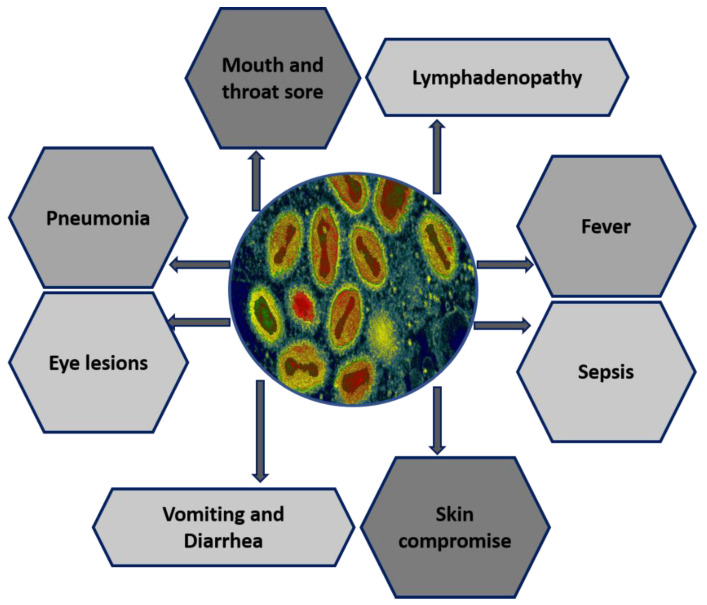
Multiorgan system involvement of MPXV.

**Figure 2 biomedicines-11-01832-f002:**
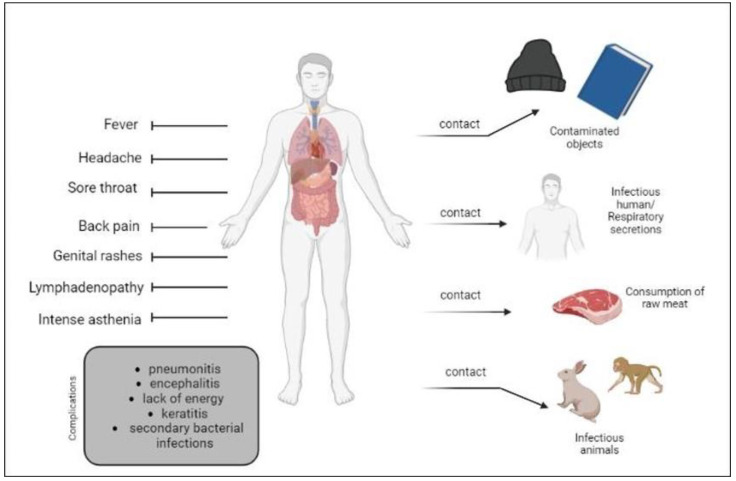
Schematic illustration of the transmission and clinical characteristics of Mpox.

**Figure 3 biomedicines-11-01832-f003:**
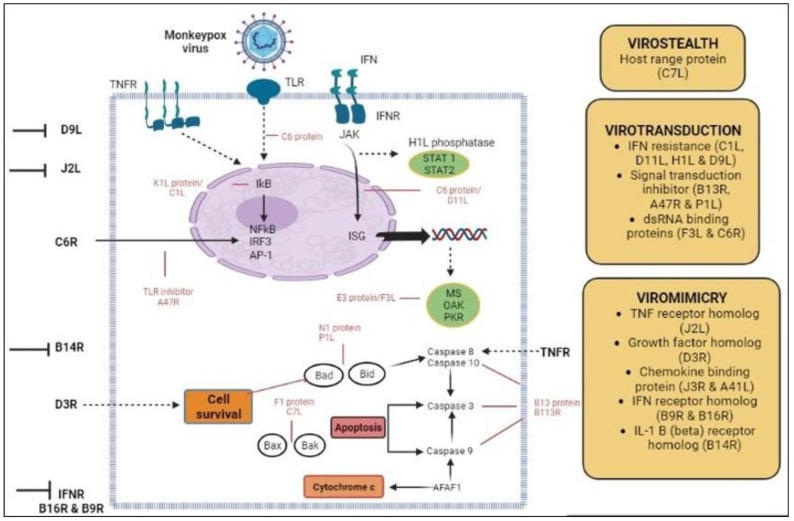
The MPXV proteins (red) that participate in virotransduction, viromimicry, and virostealth are responsible for immune evasion mechanisms of Mpox infection.

**Figure 4 biomedicines-11-01832-f004:**
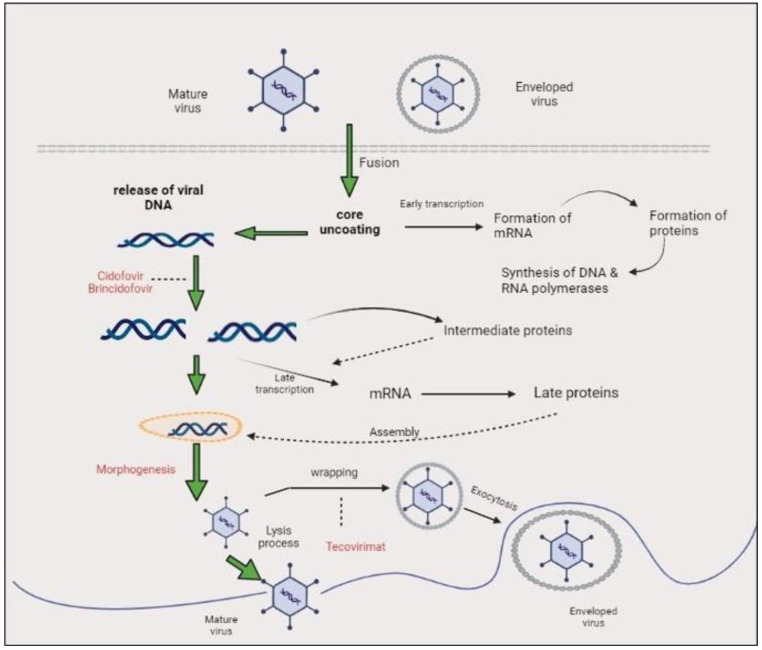
Monkeypox virus life cycle and mechanisms of action of antivirals. This diagram depicts the life cycle of MPXV inside a human cell. Notably, replication cycle of MPXV occurs in the cytoplasm of the host cell. Following viral attachment, virion binds and fuses with the host cell membrane, and the viral core is released into the cytoplasm of the host cell. Viral particles are assembled into intracellular mature viruses (IMVs), then stay in the cytoplasm as IMVs and are released as extracellular enveloped viruses during cell lysis. Mature virus can also wrap an additional envelope and attach to the cell membrane, then be released through exocytosis. Cidofovir and its prodrug brincidofovir inhibit the viral DNA polymerase during DNA replication. Tecovirimat targets the VP37 protein, which is vital for envelopment of IMV with Golgi-derived membrane to form extracellular enveloped virus (EEV), prevents the virus from leaving an infected cell, hindering the spread of the virus within the body.

**Figure 5 biomedicines-11-01832-f005:**
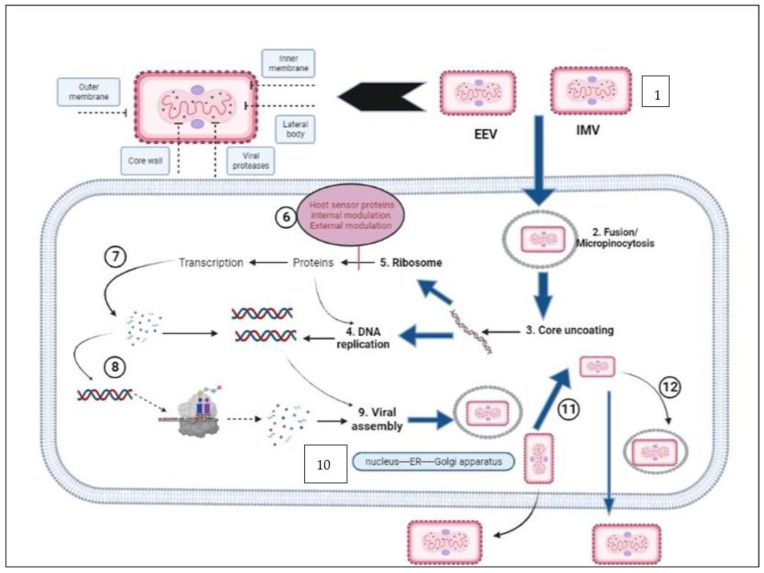
Steps of MPXV entry into host cells. (1) Schematic of the structure of MPXV. (2) Both the EEV and IMV virions penetrate the host membrane by binding and macropinocytosis. MPXV virions use glycosaminoglycans as host receptors. (3) After the internal virion components enter the cytoplasm, core uncoating occurs, and this process leads to delivery of the MPXV genome and accessory proteins to the cytosol. (4) The released MPXV genome is used as a template for DNA replication. (5) Early viral DNA transcription followed by translation into the host ribosome occurs to encode essential proteins. Early proteins aid in DNA replication. (6) These proteins interact with host sensor proteins, resulting in internal and external modulations. The major intracellular modulations include prevention of viral genome detection, induction of cell cycle arrest, apoptosis inhibition, inhibition of the antiviral system, and modulation of some host cellular signaling pathways. Early proteins play essential extracellular roles as immunomodulatory agents and as growth-factor-like domains that stimulate onset of mitosis in neighboring cells. (7) Early proteins are used in production of intermediate proteins. (8) These proteins are involved in late transcription and translation processes and aid in DNA replication. (9) Late proteins are essential components for viral assembly. (10) Viral morphogenesis occurs by formation of inner tubular nucleocapsid structure folding and assembly of viral glycoproteins to generate IMV virions. (11) Except those released via infected cell lysis, IMV virions transit to the Golgi apparatus along microtubules for double membrane wrapping. (12) The resulting EEV virions exit the infected cell by two routes: by the actin tail assembly, which provides enough force to propel the virions out of the cell, or by budding from a cellular membrane. EEV: extracellular enveloped virus. IMV: intracellular mature virus.

**Figure 6 biomedicines-11-01832-f006:**
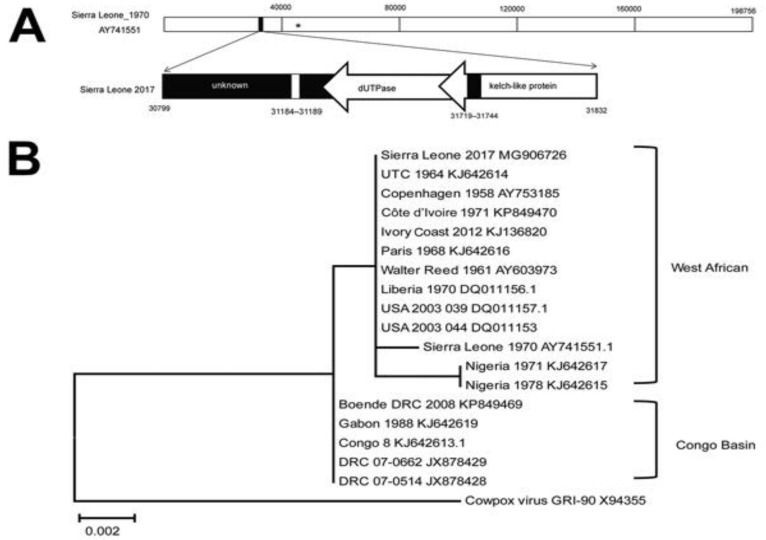
Phylogenetic analysis and molecular signature of MPXV [134]. Phylogenetic study and comparative molecular signature of the previously collected MPXV isolates and MPXV isolated in Sierra Leone in 2017 [134]. (**A**) A graphical representation of the genomic fragment of MPXV in Sierra Leone, 2017, with reference to the data on genome of MPXV Sierra Leone 1970. There are three sections that make up the MPXV Sierra Leone 2017 virus: an unknown area, genes that encode dUTPase, and genes that encode a partial kelch-like protein. The binding site of primers that are utilized for real-time PCR detection. Bottom panel displays genes reported. The direction of transcription is shown by the arrows. (**B**) Phylogenetic relationship between genomic segments of other OPXVs and MPXV collected in Sierra Leone. Neighbor-joining phylograms built with the maximum-likelihood approach with the software “MEGA version 6.0” (https://www.megasoftware.net). The scale bar represents the number of nucleotide changes at each location.

**Table 1 biomedicines-11-01832-t001:** The diagnostic techniques for MPXV among its natural hosts.

Animals Infected by Mpox	Geographical Location	Detection Technique	References
Gambian pouched rats	African territory	Viral isolation and PCR	[50]
Monkeys (*Sooty mangabey*)	Côte d’Ivoire	Molecular-testing PCR	[51]
Macaques (Cynomolgus)	Singapore/Copenhagen	Isolation of virus	[52]
Macaques (Rhesus)	Copenhagen	Antibody testing	[53]
Opossums	South America	Viral isolation and PCR	[54]
Monkeys (Asian)	Copenhagen	Isolation of virus	[55]
Hedgehogs (African)	Africa	Viral isolation, PCR, and antibody detection	[55]
Sun squirrels	Zaire	Detection of antibody	[56]
Woodchucks	USA	Viral isolation and PCR	
Jerboas	Illinois, USA	Viral isolation, PCR, and antibody detection	
Shot-tailed opossums	USA	Viral isolation, PCR, and antibody detection	
Giant anteaters (*Myrmecophaga tridactyla*)	Rotterdam	Isolation of virus	[57]
Porcupines (*Atherurus africanus*)	Zaire	Viral isolation and PCR	
Elephant shrews	DRC	Serological test	
Prairie dogs	USA	Viral isolation and PCR	
Rope squirrels	Zaire	Viral isolation and PCR	
Domestic pigs	DRC	Serological test	
Dormice (African)	USA	Viral isolation and PCR	

**Table 2 biomedicines-11-01832-t002:** Outbreaks history (epidemiology of human Mpox outbreaks [83].

Country	Duration	Cases Reported	Mortality
DRC	1970	1	100%
DRC	1981–1986	338	9.8%
DRC	1996–1997	92	3.3%
DRC	2001–2013	17,186	2.46%
DRC	2017	88	6.3%
Sudan	2005	37	0
Cameron	1989	1	0
Nigeria	1971	2	0
Nigeria	1978	1	0
Nigeria	2017–2018	228	2.6%
Gabon	1991	9	0
Sierra Leone	1970–1971	4	0
USA	2003	47	0
Central African Republic	2015–2018	104	8.3%

**Table 3 biomedicines-11-01832-t003:** Clinical picture comparison.

Character	Smallpox	Monkeypox	Varicella	References
Viral cycle completion	28 days	28 days	21 days	[87,89,95]
Incubation time	2 weeks	2 weeks	3 weeks	
Lesion inflammatory cycle	2–4 weeks	2–4 weeks	1–3 weeks	
Body temperature	>40 °C	38.5–40.5 °C	38 ± 8 °C	
Lymphadenopathy	Often	Often	Rare	
Lesion	Centrifugal	Centrifugal	Centripetal	

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
