# Peer review of "An Updated Review on Monkeypox Viral Disease: Emphasis on Genomic Diversity"

_biomedicines, 2023, doi:10.3390/biomedicines11071832_

Round 1

Reviewer 1 Report

In their review, the authors focus on the genomic diversity of monkeypox.

This is an extremely important topic with high relevance for scientists and physicians worldwide. I had great expectations of the article, which were at least partially disappointed. 

I must inform the authors that the article is altogether very superficial and urgently needs a comprehensive revision. It should go into more detail, which is of extreme importance to the readers of such a prestigious journal. 

Furthermore, the illustrations are superficial and of modest quality in terms of content and graphics.

In summary, this is a review on an extremely important topic that is in urgent need of comprehensive revisions. I strongly encourage the authors to do so and am already looking forward to reading the revised version. 

Moderate editing of English language recommended

Author Response

Reviewer 1

Comments and Suggestions for Authors

In their review, the authors focus on the genomic diversity of monkeypox.

This is an extremely important topic with high relevance for scientists and physicians worldwide. I had great expectations of the article, which were at least partially disappointed. I must inform the authors that the article is altogether very superficial and urgently needs a comprehensive revision. It should go into more detail, which is of extreme importance to the readers of such a prestigious journal. In summary, this is a review on an extremely important topic that is in urgent need of comprehensive revisions. I strongly encourage the authors to do so and am already looking forward to reading the revised version. 

Response: Dear reviewer, thank you for your valuable comments and suggestions. The manuscript has been thoroughly revised according to the comments from you and other reviewers. We must appreciate that, after addressing comments from you and other reviewers, the quality has been increased significantly. Furthermore, we have revised the manuscript for English proofreading and grammatical mistakes and more literature has been added to increase the scientific soundness of the manuscript.

Following literature has been added: Line 78-83, 106-107, 121-126, 185-190, 231-233, 249-253, 264-268, 280-286, 296-306, 329-348, 409-420, figure 4 and 6.

Furthermore, the illustrations are superficial and of modest quality in terms of content and graphics.

Response: The qualities of the figures has been improved. Also 2 new figures have been added (Figure 4 and 6). More description has been added to figure 4, 5 and 6.

Comments on the Quality of English Language

Moderate editing of English language recommended

Response: The manuscript has been thoroughly revised for English proofreading and grammatical mistakes. Furthermore, more literature has been added to increase the scientific soundness of the manuscript.

Reviewer 2 Report

This is an interesting manuscript and is an up-to-date overview on genomic diversity of Monkeypox, and points to the importance of monitoring cases in order to understand the epidemiology of the disease and consequent risk of new pandemy. 

References are updated.

Some suggestions:

MPXV, according to the first appearing, means monkeypox virus, but sometimes appears as the disease.

Figures, mainly 3 and 4, need an improved resolution

The text need some minor review. For example, line 100 and 101 bring "can" and "could" in the same phrase. Rewrite with only one.

Line 83 ...can be transfered

Author Response

Reviewer 2

Comments and Suggestions for Authors

This is an interesting manuscript and is an up-to-date overview on genomic diversity of Monkeypox, and points to the importance of monitoring cases in order to understand the epidemiology of the disease and consequent risk of new pandemy. References are updated.

Response: Dear reviewer, thank you for your valuable comments and suggestions. The manuscript has been thoroughly revised according to the comments from you and other reviewers. We must appreciate that, after addressing comments from you and other reviewers, the quality has been increased significantly. Furthermore, we have revised the manuscript for English proofreading and grammatical mistakes and more literature has been added to increase the scientific soundness of the manuscript.

Some suggestions:

MPXV, according to the first appearing, means monkeypox virus, but sometimes appears as the disease.

Response: Dear reviewer, we have rechecked for the differences between virus and the disease. Corrections has been made. (Line 86, 98, 99, 219, 272, 455)

Figures, mainly 3 and 4, need an improved resolution

Response: The qualities of the figures has been improved. Also 2 new figures have been added (Figure 4 and 6).

Comments on the Quality of English Language

The text need some minor review. For example, line 100 and 101 bring "can" and "could" in the same phrase. Rewrite with only one.

Response: Line 106-107: the sentence has been corrected. The manuscript has been thoroughly revised for English proofreading and grammatical mistakes. Furthermore, more literature has been added to increase the scientific soundness of the manuscript.

Line 83 ...can be transfered

Response: Line 89: the sentence has been corrected.

Reviewer 3 Report

Abstract

Line 60 – The official name of Republic of Congo is Democratic Republic of Congo (DRC) that is a country of Central Africa… so the disease “…had spread over West and other Central African countries.” 

1.       Introduction

Lines 78-80 – There is three clades not two (Congo Basin clade – clade I; Western African clades – clade IIa and clade IIb). These clades are also nominated variants.

Lines 81-83 – The first human’s case was described by Ladnyj ID et al “A human infection caused by monkeypox virus in Basankusu Territory, Democratic Republic of Congo”. Bull OrgMond Santé 1972;46:593-597

Lines 88-89 – The same that is stated in lines 81-83. So avoid a repeat.

Figure 1 and figure 2 – the figures are originals or reproductions?

Lines 113-119 – A clinical sign differentiating monkeypox from smallpox is lymphadenopathy. On the other hand, the new mpox outbreak clinical presentation has been atypical as compared to previously document reports in mpox endemic areas of Africa. This point is important taking in account mpox clinical presentation as described (for instance “high grade fever” and “vesicular pustule eruption abundantly present on the face, hands, palm, feet and soles or minimum 5 smallpox scabs”). However, few lesions, no lesions, but anal pain and bleeding, genital or perineal/perianal alone lesion, and absence of prodromal period or constitutional symptoms appearing after the lesion are features of the new mpox outbreak.

Lines 124-126 – Since 2017, there is a large outbreak of mpox in Nigeria, linked with cases on 2018-2019 outside endemic areas of Africa and probably with the new outbreak starting on May 2022.

Lines 126-127 – The 2003 mpox outbreak in United States was associated with shipment of approximately 800 small mammals from Ghana that contained 762 African rodents, including squirrels, rats, dormice’s, and porcupines, so it’s not correct “…through a pet trade of rat shipment.” 

2.       Progressive epidemiology of Monkeypox virus

Lines 164-166 – In 2008, two separately imported cases of human mpox in United Kingdom (UK) were reported. Both had travelled in southern Nigeria before coming to UK [Vaughan A et al. Eurosurveillance 2018;23(38):pii=1800509]. So these two cases must be added to the cases diagnosed in Singapore and Israel.

Lines 167-169 – “The 183 cases that were reported by Nigeria CDC do not include the three cases that occurred in the United Kingdom and originated in Nigeria [59]”. The reference 59, Taseen S et al. Journal of Medical Virology 2022;95:e27948, do not have any data on “…the three cases that occurred in the United Kingdom…”.

Lines 169-171 – “The Congo DRC”, which is the country? DRC abbreviation means Democratic Republic of Congo, that is the country with the most number of incidence of human mpox. 

3.       Monkeypox and other orthopoxviruses

Lines 180-182 – Airborne transmission route must be considered as a possible transmission mode under the conditions of current experimental and analytical findings. The findings and the analysis with aerosol dynamics show that aerosols carrying MPXV could be present in environments where patients have resided and that airborne transmission of MPXV can occur (Gould S et al. Lancet Microbe 2022;3: e904-11; Hernaez B et al. Lancet Microbe 2023;4:e21-28).

Lines 190-191 – “Variola, a variation of variola with a rate…” what does it means?

Lines 195-196 – There are three genetically separates virus clades, not two.

Lines 199-201 – “Congo DRC”. What does it mean? In Central Africa there is two separate countries, one the Democratic Republic of the Congo (DRC) and the other Republic of the Congo.

Cases of mpox in Africa were comparatively small before the 1980s, following smallpox vaccination discontinuation worldwide, and in West and Central Africa the risk of human mpox outbreaks has been growing every year since that date. The smallpox vaccination provides cross-immunity against mpox. Since 1977, the end of smallpox vaccination resulted in a decrease immunity, and in an increase susceptible population to mpox. from the smallpox vaccination. That’s explain why mpox was rare before 1977, smallpox eradicated, and thereafter the increase of mpox transmission. 

6.       Genetic clades of monkeypox

The title should be “Clades of monkeypox”. In virology, viruses are placed in clades based on phylogenetic trees constructed from their genome sequences. The viral clades share similar genetic sequences.

Please update with several articles with the reference to the three clades of mpox virus that are now recognized.

Author Response

Reviewer 3

Comments and Suggestions for Authors

Dear reviewer, thank you for your valuable comments and suggestions. The manuscript has been thoroughly revised according to the comments from you and other reviewers. We must appreciate that, after addressing comments from you and other reviewers, the quality has been increased significantly. Furthermore, we have revised the manuscript for English proofreading and grammatical mistakes and more literature has been added to increase the scientific soundness of the manuscript.

Abstract

Line 60 – The official name of Republic of Congo is Democratic Republic of Congo (DRC) that is a country of Central Africa… so the disease “…had spread over West and other Central African countries.” 

Response: Line 60: the country name has been corrected.

  1. Introduction

Lines 78-80 – There is three clades not two (Congo Basin clade – clade I; Western African clades – clade IIa and clade IIb). These clades are also nominated variants.

Response: Line 84-85: the mistake has been corrected.

Lines 81-83 – The first human’s case was described by Ladnyj ID et al “A human infection caused by monkeypox virus in Basankusu Territory, Democratic Republic of Congo”. Bull OrgMond Santé 1972;46:593-597

Response: Line 87-88: The sentence has been revised and the reference has been cited.

Lines 88-89 – The same that is stated in lines 81-83. So avoid a repeat.

Response: Line 94: The repetition has been removed.

Figure 1 and figure 2 – the figures are originals or reproductions?

Response: Dear reviewer, all of the figures in the current manuscript are originally made and no need for the copyright permission or any other ethical considerations.

Lines 113-119 – A clinical sign differentiating monkeypox from smallpox is lymphadenopathy. On the other hand, the new mpox outbreak clinical presentation has been atypical as compared to previously document reports in mpox endemic areas of Africa. This point is important taking in account mpox clinical presentation as described (for instance “high grade fever” and “vesicular pustule eruption abundantly present on the face, hands, palm, feet and soles or minimum 5 smallpox scabs”). However, few lesions, no lesions, but anal pain and bleeding, genital or perineal/perianal alone lesion, and absence of prodromal period or constitutional symptoms appearing after the lesion are features of the new mpox outbreak.

Response: Line 121-126: Dear reviewer, thank you for your valuable suggestions. We have revised the paragraph and added new information as suggested.

Lines 124-126 – Since 2017, there is a large outbreak of mpox in Nigeria, linked with cases on 2018-2019 outside endemic areas of Africa and probably with the new outbreak starting on May 2022.

Response: Line 185-187: New information has been added and the reference has been cited.

Lines 126-127 – The 2003 mpox outbreak in United States was associated with shipment of approximately 800 small mammals from Ghana that contained 762 African rodents, including squirrels, rats, dormice’s, and porcupines, so it’s not correct “…through a pet trade of rat shipment.” 

Response: Line 188-190: The sentence has been revised and a reference has been cited.

  1. Progressive epidemiology of Monkeypox virus

Lines 164-166 – In 2008, two separately imported cases of human mpox in United Kingdom (UK) were reported. Both had travelled in southern Nigeria before coming to UK [Vaughan A et al. Eurosurveillance 2018;23(38):pii=1800509]. So these two cases must be added to the cases diagnosed in Singapore and Israel.

Response: Line 231-233: New information has been added and the reference has been cited.

Lines 167-169 – “The 183 cases that were reported by Nigeria CDC do not include the three cases that occurred in the United Kingdom and originated in Nigeria [59]”. The reference 59, Taseen S et al. Journal of Medical Virology 2022;95:e27948, do not have any data on “…the three cases that occurred in the United Kingdom…”.

Response: Line 231: Correct reference has been cited.

Lines 169-171 – “The Congo DRC”, which is the country? DRC abbreviation means Democratic Republic of Congo, that is the country with the most number of incidence of human mpox. 

Response: Line 233: The “congo” has been removed.

  1. Monkeypox and other orthopoxviruses

Lines 180-182 – Airborne transmission route must be considered as a possible transmission mode under the conditions of current experimental and analytical findings. The findings and the analysis with aerosol dynamics show that aerosols carrying MPXV could be present in environments where patients have resided and that airborne transmission of MPXV can occur (Gould S et al. Lancet Microbe 2022;3: e904-11; Hernaez B et al. Lancet Microbe 2023;4:e21-28).

Response: Line 249-253: Dear reviewer, thank you for your valuable suggestion. We have added the relevant information and cited the respective references to strengthen the statements.

Lines 190-191 – “Variola, a variation of variola with a rate…” what does it means?

Response: Dear reviewer, the respective sentence has been removed from the revised version of manuscript as it was probably a repetition from above sentence (Line 239).

Lines 195-196 – There are three genetically separates virus clades, not two.

Response: Line 262: The correction has been made.

Lines 199-201 – “Congo DRC”. What does it mean? In Central Africa there is two separate countries, one the Democratic Republic of the Congo (DRC) and the other Republic of the Congo.

Response: Line 269: The “congo” has been removed.

Cases of mpox in Africa were comparatively small before the 1980s, following smallpox vaccination discontinuation worldwide, and in West and Central Africa the risk of human mpox outbreaks has been growing every year since that date. The smallpox vaccination provides cross-immunity against mpox. Since 1977, the end of smallpox vaccination resulted in a decrease immunity, and in an increase susceptible population to mpox. from the smallpox vaccination. That’s explain why mpox was rare before 1977, smallpox eradicated, and thereafter the increase of mpox transmission. 

Response: Dear reviewer, thank you so much for the very nice explanation. We have revised the paragraph accordingly (Line 280-286)

  1. Genetic clades of monkeypox

The title should be “Clades of monkeypox”. In virology, viruses are placed in clades based on phylogenetic trees constructed from their genome sequences. The viral clades share similar genetic sequences.

Response: Line 357: The heading has been changed as suggested.

Please update with several articles with the reference to the three clades of mpox virus that are now recognized.

Response: The references has been updated and more latest references has been cited to strengthen the statements.

Reviewer 4 Report

The review article submitted by Rabaan et al on Biomedicines on the Monkey pox viral disease is a good attempt by the authors in putting together the available data on the genetic diversity. However, the review is poorly organized and therefore the reader is unclear or confused in many areas.

The authors must reframe the review in four main sections like (A) Geographical range and epidemiology (B) Genome organization (C) replication cycle (D) Phylogenetics-  in my opinion so that they can avoid repeating the facts again and again in different sections. For eg: Line 195-199: “The virus is ……..western Africa “. This is presented in the first paragraph in line 81 and again repeated in Line 88. Many other facts are also repeated like this throughout the review and hence sounds very distracting.

The members of the family needs to be mentioned in the beginning. Otherwise the reader won’t understand why you mentioned the name of new viruses in some sections without referring them before. For eg: the geographical spread of cow pox in Line 130 is mentioned without mentioning it anywhere before.

Section 3 Monkeypox and other orthopoxviruses: This subtitle is confusing and the subject pertaining to this section is how the epidemiology of other pox viruses and how the vaccination of others affect the epidemiology of MPXV.

Give an additional column to all tables (Tables1-4) for citing references.

Include a schematic representation of the genome organization of monkey pox and describe it and then explain the replication cycle.

Figure 3 and the replication cycle text (Line 225-245)is not well explained. Few steps like step 1, 7,8,9,10, 11, 12 are not explained in the text or in figure legend while just numbered. In the text few acronyms are mentioned IMV, EEV which is not expanded. Please be careful in using abbreviations for the first time and try to expand it or explain it.

I feel the section genetic variability should come the very first- so that the reader can relate the members of the family

Even though the authors are trying to explain the genetic variability- I am not seeing any phylogenetic diagram to explain the genetic diversity of the isolated monkey pox virus which is very critical for the article. And I don’t understand what is A and B mentioned in the section 7. If you are referring to other phylogenetics study that needs to be interpreted and presented in a different style. Basically the section 7 is very difficult to understand without a diagram or its not informative

Language usage is fine. But the artcle is not very precise to the subject and therefore needs to be written more scientifically.

Author Response

Reviewer 4

Comments and Suggestions for Authors

The review article submitted by Rabaan et al on Biomedicines on the Monkey pox viral disease is a good attempt by the authors in putting together the available data on the genetic diversity. However, the review is poorly organized and therefore the reader is unclear or confused in many areas.

Response: Dear reviewer, thank you for your valuable comments and suggestions. The manuscript has been thoroughly revised according to the comments from you and other reviewers. We must appreciate that, after addressing comments from you and other reviewers, the quality has been increased significantly. Furthermore, we have revised the manuscript for English proofreading and grammatical mistakes and more literature has been added to increase the scientific soundness of the manuscript.

The authors must reframe the review in four main sections like (A) Geographical range and epidemiology (B) Genome organization (C) replication cycle (D) Phylogenetics-  in my opinion so that they can avoid repeating the facts again and again in different sections. For eg: Line 195-199: “The virus is ……..western Africa “. This is presented in the first paragraph in line 81 and again repeated in Line 88. Many other facts are also repeated like this throughout the review and hence sounds very distracting.

Response: Dear reviewer, thank you for your valuable suggestion to rearrange the headings and literature. The information has been rearranged as per your suggestion. Furthermore, we have thoroughly revised the manuscript for any repetitions of sentences.

The members of the family needs to be mentioned in the beginning. Otherwise the reader won’t understand why you mentioned the name of new viruses in some sections without referring them before. For eg: the geographical spread of cow pox in Line 130 is mentioned without mentioning it anywhere before.

Response: Line 78-83: New information has been added and relevant references has been cited.

Section 3 Monkeypox and other orthopoxviruses: This subtitle is confusing and the subject pertaining to this section is how the epidemiology of other pox viruses and how the vaccination of others affect the epidemiology of MPXV.

Response: The subsection 3 has been removed. And new information has been added at line 264-268. The new information will be more understandable for the reader.

Give an additional column to all tables (Tables1-4) for citing references.

Response: References in table 1 and 3 has been added. Reference for table 2 already provided. There is no table 4.

Include a schematic representation of the genome organization of monkey pox and describe it and then explain the replication cycle.

Response: Figure 4 has been newly added. (Line 296-306)

Figure 3 and the replication cycle text (Line 225-245) is not well explained. Few steps like step 1, 7,8,9,10, 11, 12 are not explained in the text or in figure legend while just numbered. In the text few acronyms are mentioned IMV, EEV which is not expanded. Please be careful in using abbreviations for the first time and try to expand it or explain it.

Response: Figure 3 has been changed as figure 5. The explanation for steps has been added. The abbreviations have been added.

I feel the section genetic variability should come the very first- so that the reader can relate the members of the family

Response: We appreciate you suggestion. We have moved the genetic variability section as section 1.1 under the introduction.

Even though the authors are trying to explain the genetic variability- I am not seeing any phylogenetic diagram to explain the genetic diversity of the isolated monkey pox virus which is very critical for the article. And I don’t understand what is A and B mentioned in the section 7. If you are referring to other phylogenetics study that needs to be interpreted and presented in a different style. Basically, the section 7 is very difficult to understand without a diagram or its not informative

Response: Line 409-420: Dear reviewer, thank you for your valuable suggestion to add a phylogenetic analysis from previous study. Actually, this figure was already there previously but somehow was removed. The information in A and B was belonging to the figure. The respective reference has been cited. (Figure 6)

Comments on the Quality of English Language

Language usage is fine. But the artcle is not very precise to the subject and therefore needs to be written more scientifically.

Response: The manuscript has been thoroughly revised for English proofreading and grammatical mistakes. Furthermore, more literature has been added to increase the scientific soundness of the manuscript.

Round 2

Reviewer 1 Report

Congratulation to this excellent revised version. The review process resulted in a significantly improved manuscript with extremely high relevance. All issues have been addressed comprehensively.

Author Response

Reviewer 1

Comments and Suggestions for Authors

Congratulation to this excellent revised version. The review process resulted in a significantly improved manuscript with extremely high relevance. All issues have been addressed comprehensively.

Response: Dear reviewer, we would like to really appreciate your kind efforts for providing us with your expert opinions, suggestions to our manuscript. The quality of manuscript has been significantly improved after addressing the comments from your side and other reviewers.

Reviewer 3 Report

Consider the review of the title to “An updated review on mpox: Emphasis on genomic diversity”. Since WHO, in November 2022, announced the adoption of mpox as the preferred term for monkeypox. The virus is currently still termed Monkeypox virus. So correct in the body of the text and titles of figures and tables monkeypox to mpox. 

Abstract

Line 60 – Please change the name of the country to Democratic Republic of the Congo (DRC).

Line 63 – Please correct to Monkeypox virus (MPOX).

1.       Introduction

Line 79 – Orthopoxvirus (OPVX)

Line 86 – Please correct to… causes mpox was first…

Line 88 – Please correct to Democratic Republic of the Congo (DRC) described...

Line 92 – Please correct to… for mpox lasts 6-13 days, however…

Line 96 – It’s not a error to start a sentence with a figure, but it often looks award. You should either reword your sentence or write the year 2003 in full.

Line 98 – Please correct to Mpox got the ability…

Line 99 – Please correct to… new reservoirs of mpox to be…

Figure 1 – Please correct to Multiorgan system involvement of MPXV. Use the same abbreviation as in the body of the article.

Line 120 – In body text, spell out whole numbers from zero through nine, and use numerals for 10 or greater. It’s OK to use numerals for zero to nine who you have limited spaces, such as in tables and UI. So correct to… or minimum five small scabs [29,30].

Lines 138, 144, 150, 159, 227, 307, 314, 318, 324, 344, and 418 – What means ORPs, CRS, ITRs, OMCP, NISG2D, CDC, IMV, EEV, IEVs, TGN, ATIs, MV, and MEGA6. Acronyms and abbreviations must be spelled out the full phrase or term the first time you use it in your paper and include the abbreviation in parentheses. You must use the abbreviation each time after that, and not spell out again.

Line 139 – Acronyms and abbreviations must be spelled out completely on the begin a sentence. So correct to Cowpox virus only causes…

Lines 163-164 – Abbreviations should only be used if the organization or term appears two or more times in the text. So “Amplicon Sanger-sequencing product… 

2.       Geographical range and progressive epidemiology of MPXV

Table 1 – Please correct DR Congo to DRC.

Line 203 – Please correct to… was just four [60].

Line 206 – Please correct to… 8.7%.

Line 210 – Please correct to… particularly in DRC where…

Line 218 – Please correct to… in DRC [70,71]. The DRC…

Line 219 – Please correct to… incidence of mpox, and no other…

Line 225 – Please correct to… for the DRC.

Table 2 – Use DRC instead Congo and Central African Republic instead of Central Africa.

Line 255 – Please correct to… ranging from 10% to 30% have been…

Line 272 – Please correct to… contact mpox cases…

Line 274 – Please correct to… ranged from 37% to 88% [103].

Line 279 – Please correct to… younger than five years old… 

3.       Genome organization, replication cycle and morphology

Lines 292-293 – Please correct to… repeats ORFs which combined…

Line 296 (Figure 4) – Please correct to Monkeypox virus life cycle…

Line 297 (Figure 4) – Please correct to Notably, replication cycle of MPXV occurs…

Line 301 (Figure 4) – Please correct to Mature virus can also wrap…

Line 311 – Please correct to… shared by all OPXV.

Line 316 – Please correct to Cell associated virions are the primary…

Lines 322-323 – Please correct to… of dense particles that are not…

Line 325 – Please correct to…strains of CPXV, in which…

Lines 373-374 – Please correct to… and related OPXV have shown… 

4.       Phylogenetic analysis or MPXV

Line 380 – Please correct to… and only 95% nucleic acid…

Line 384 – Please correct to… first two cycles of viral…

Line 388 – Please correct to… using the four different…

Lines 389-390 – Please correct to… embedded with VACV.

Line 392 – Please correct to Comparison of genome sequence…

Line 393 – Please correct to… to identify two target genomic region (R) positioned…

Line 417 – Please correct to… OPXV and MPXV collected in Sierra Leone

Lines 428-430 – Please correct to… related to the OPXV genus… and to… OPXV exhibit significant…

Line 432 – Please correct to Nigerian five isolates…

Line 441 – Please correct to Furthermore, the West African clade…

Lines 448-449 – Please correct to… and Gabon makeup the Congo Bassin clade; and… MPXV resolved in the Congo Bassin clade… 

5.       Conclusion

Line 455 – Please correct to… age of those with mpox, as well…

Author Response

Reviewer 3

Comments and Suggestions for Authors

Consider the review of the title to “An updated review on mpox: Emphasis on genomic diversity”. Since WHO, in November 2022, announced the adoption of mpox as the preferred term for monkeypox. The virus is currently still termed Monkeypox virus. So correct in the body of the text and titles of figures and tables monkeypox to mpox. 

Response: Dear reviewer, we would like to really appreciate your kind efforts for providing us with your expert opinions, suggestions to our manuscript. The quality of manuscript has been significantly improved after addressing the comments from your side and other reviewers. The abbreviated forms have been corrected throughout the manuscript.

Abstract

Line 60 – Please change the name of the country to Democratic Republic of the Congo (DRC).

Response: Line 60: The country name has been corrected.

Line 63 – Please correct to Monkeypox virus (MPOX).

Response: Line 59, 63, 64 and 68 : Corrected.

  1. Introduction

Line 79 – Orthopoxvirus (OPVX)

Response: Line 79: Corrected.

Line 86 – Please correct to… causes mpox was first…

Response: Line 86: Corrected.

Line 88 – Please correct to Democratic Republic of the Congo (DRC) described...

Response: Line 87-88: The country name has been corrected.

Line 92 – Please correct to… for mpox lasts 6-13 days, however…

Response: Line 91: Corrected.

Line 96 – It’s not a error to start a sentence with a figure, but it often looks award. You should either reword your sentence or write the year 2003 in full.

Response: Line 96: The sentence has been corrected.

Line 98 – Please correct to Mpox got the ability…

Response: Line 98: Corrected.

Line 99 – Please correct to… new reservoirs of mpox to be…

Response: Line 99: Corrected.

Figure 1 – Please correct to Multiorgan system involvement of MPXV. Use the same abbreviation as in the body of the article.

Response: Line 117: Corrected.

Line 120 – In body text, spell out whole numbers from zero through nine, and use numerals for 10 or greater. It’s OK to use numerals for zero to nine who you have limited spaces, such as in tables and UI. So correct to… or minimum five small scabs [29,30].

Response: Line 120: Corrected.

Lines 138, 144, 150, 159, 227, 307, 314, 318, 324, 344, and 418 – What means ORPs, CRS, ITRs, OMCP, NISG2D, CDC, IMV, EEV, IEVs, TGN, ATIs, MV, and MEGA6. Acronyms and abbreviations must be spelled out the full phrase or term the first time you use it in your paper and include the abbreviation in parentheses. You must use the abbreviation each time after that, and not spell out again.

Response: The abbreviations have been written at their first appearance. E.g., Line 160, 149, 137, 159, 226.

Line 139 – Acronyms and abbreviations must be spelled out completely on the begin a sentence. So correct to Cowpox virus only causes…

Response: Line 139: Corrected.

Lines 163-164 – Abbreviations should only be used if the organization or term appears two or more times in the text. So “Amplicon Sanger-sequencing product… 

Response: Line 164: Corrected.

  1. Geographical range and progressive epidemiology of MPXV

Table 1 – Please correct DR Congo to DRC.

Response: Table 1: Corrected.

Line 203 – Please correct to… was just four [60].

Response: Line 202: Corrected.

Line 206 – Please correct to… 8.7%.

Response: Line 203, 205: Corrected.

Line 210 – Please correct to… particularly in DRC where…

Response: Line 209: Corrected.

Line 218 – Please correct to… in DRC [70,71]. The DRC…

Response: Line 217: Corrected.

Line 219 – Please correct to… incidence of mpox, and no other…

Response: Line 218: Corrected.

Line 225 – Please correct to… for the DRC.

Response: Line 223: Corrected.

Table 2 – Use DRC instead Congo and Central African Republic instead of Central Africa.

Response: Table 2: Corrected.

Line 255 – Please correct to… ranging from 10% to 30% have been…

Response: Line 253: Corrected.

Line 272 – Please correct to… contact mpox cases…

Response: Line 270: Corrected.

Line 274 – Please correct to… ranged from 37% to 88% [103].

Response: Line 272: Corrected.

Line 279 – Please correct to… younger than five years old… 

Response: Line 277: Corrected.

  1. Genome organization, replication cycle and morphology

Lines 292-293 – Please correct to… repeats ORFs which combined…

Response: Line 290-291: Corrected.

Line 296 (Figure 4) – Please correct to Monkeypox virus life cycle…

Response: Line 294: Corrected.

Line 297 (Figure 4) – Please correct to Notably, replication cycle of MPXV occurs…

Response: Line 295: Corrected.

Line 301 (Figure 4) – Please correct to Mature virus can also wrap…

Response: Line 299: Corrected.

Line 311 – Please correct to… shared by all OPXV.

Response: Line 309: Corrected.

Line 316 – Please correct to Cell associated virions are the primary…

Response: Line 314: Corrected.

Lines 322-323 – Please correct to… of dense particles that are not…

Response: Line 321: Corrected.

Line 325 – Please correct to…strains of CPXV, in which…

Response: Line 323: Corrected.

Lines 373-374 – Please correct to… and related OPXV have shown… 

Response: Line 371: Corrected.

  1. Phylogenetic analysis or MPXV

Line 380 – Please correct to… and only 95% nucleic acid…

Response: Line 377: Corrected.

Line 384 – Please correct to… first two cycles of viral…

Response: Line 381: Corrected.

Line 388 – Please correct to… using the four different…

Response: Line 385: Corrected.

Lines 389-390 – Please correct to… embedded with VACV.

Response: Line 386: Corrected.

Line 392 – Please correct to Comparison of genome sequence…

Response: Line 389: Corrected.

Line 393 – Please correct to… to identify two target genomic region (R) positioned…

Response: Line 390: Corrected.

Line 417 – Please correct to… OPXV and MPXV collected in Sierra Leone

Response: Line 414: Corrected.

Lines 428-430 – Please correct to… related to the OPXV genus… and to… OPXV exhibit significant…

Response: Line 425-426: Corrected.

Line 432 – Please correct to Nigerian five isolates…

Response: Line 429: Corrected.

Line 441 – Please correct to Furthermore, the West African clade…

Response: Line 438: Corrected.

Lines 448-449 – Please correct to… and Gabon makeup the Congo Bassin clade; and… MPXV resolved in the Congo Bassin clade… 

Response: Line 445-446-447: Corrected.

  1. Conclusion

Line 455 – Please correct to… age of those with mpox, as well…

Response: Line 452: Corrected.

Reviewer 4 Report

Thank you for addressing my concerns. I have one more suggestion. The figure 4 is not a very important one and the details of antivirals can be included in the figure 5 and remove figure 4. 

Author Response

Reviewer 4

Comments and Suggestions for Authors

Thank you for addressing my concerns. I have one more suggestion. The figure 4 is not a very important one and the details of antivirals can be included in the figure 5 and remove figure 4. 

Response: Dear reviewer, we would like to really appreciate your kind efforts for providing us with your expert opinions, suggestions to our manuscript. The quality of manuscript has been significantly improved after addressing the comments from your side and other reviewers. Thank you for your suggestion to merge figure 4 and 5 as one figure. We really appreciate this suggestion but unfortunately after merging these two figures as one, it becomes so clumpy. Hence, we would like to keep the both figures as it is. Please allow us.

Thank you!

Round 3

Reviewer 3 Report

The quality of the manuscript was improved, however a few gaps remained.

Lines 91-93 – please correct to …lasts 6-13 days, however this time frame can last anywhere 5-21 days [12-14]

Lines 133,141,144, and 174 – please use the abbreviation OPXVs

Line 161 – please correct to …this loss in six more…

Lines 192 and 209 – please use the abbreviation CPXV

Line 226 – please correct to …the Center for Disease Control (CDC)…

Line 267 – please correct to …Central and Western Africa…

Figure 4:

Line 294 – please correct to …cycle of MPXV…

Line 297 – please correct to …in the cytoplasma as IMVs…

Line 301 – please correct to …for envelopment of IMV…

Line 302 – please correct to …membrane to form extracellular enveloped virus (EEV), prevents…

Line 304 – please correct to …breakdown and EEV is released…

Line 376 – please uniform the names of regions they appear in the text. The genomic sequences of Congo Basin and West Africa derived strains…

Line 403 – Figure 6 in bold

Line 409 – please correct to …MPXV Sierra Leone 1970. There are three…

Line 423 – please correct to …less 50 genes exhibit…

Line 424 – please correct to …One hundred seventy four genes…

Author Response

Reviewer 3

Comments and Suggestions for Authors

The quality of the manuscript was improved, however a few gaps remained.

Response: Dear reviewer, thank you once again for providing us with the comments to the revied version of manuscript. We have revised the manuscript and did the changes as per your suggestions. We have highlighted all of the changes in red colour for you to follow easily.

Lines 91-93 – please correct to …lasts 6-13 days, however this time frame can last anywhere 5-21 days [12-14]

Response: Line 92: Corrected.

Lines 133,141,144, and 174 – please use the abbreviation OPXVs

Response: Line 132, 140, 143, 173: Corrected.

Line 161 – please correct to …this loss in six more…

Response: Line 159: Corrected.

Lines 192 and 209 – please use the abbreviation CPXV

Response: Line 190, 206: Corrected.

Line 226 – please correct to …the Center for Disease Control (CDC)…

Response: Line 223: Corrected.

Line 267 – please correct to …Central and Western Africa…

Response: Line 264: Corrected.

Figure 4:

Line 294 – please correct to …cycle of MPXV…

Response: Line 291: Corrected.

Line 297 – please correct to …in the cytoplasma as IMVs…

Response: Line 294: Corrected.

Line 301 – please correct to …for envelopment of IMV…

Response: Line 298: Corrected.

Line 302 – please correct to …membrane to form extracellular enveloped virus (EEV), prevents…

Response: Line 298, 301: Corrected.

Line 304 – please correct to …breakdown and EEV is released…

Response: Line 301: Corrected.

Line 376 – please uniform the names of regions they appear in the text. The genomic sequences of Congo Basin and West Africa derived strains…

Response: Line 371: Corrected.

Line 403 – Figure 6 in bold

Response: Line 398: Corrected.

Line 409 – please correct to …MPXV Sierra Leone 1970. There are three…

Response: Line 404: Corrected.

Line 423 – please correct to …less 50 genes exhibit…

Response: Line 418: Corrected.

Line 424 – please correct to …One hundred seventy four genes…

Response: Line 419: Corrected.